# A DNA Vaccine Encoding the Full-Length Spike Protein of Beta Variant (B.1.351) Elicited Broader Cross-Reactive Immune Responses against Other SARS-CoV-2 Variants

**DOI:** 10.3390/vaccines11030513

**Published:** 2023-02-22

**Authors:** Gan Zhao, Zhiyu Zhang, Yuan Ding, Jiawang Hou, Ying Liu, Mengying Zhang, Cheng Sui, Limei Wang, Xin Xu, Xiaoming Gao, Zhihua Kou

**Affiliations:** Advaccine Biopharmaceutics (Suzhou) Co., Ltd., Suzhou 215000, China

**Keywords:** SARS-CoV-2, variant of concern (VOC), cross-reactive immune responses, vaccine, DNA vaccine

## Abstract

The SARS-CoV-2 pandemic remains an ongoing threat to global health with emerging variants, especially the Omicron variant and its sub-lineages. Although large-scale vaccination worldwide has delivered outstanding achievements for COVID-19 prevention, a declining effectiveness to a different extent in emerging SARS-CoV-2 variants was observed in the vaccinated population. Vaccines eliciting broader spectrum neutralizing antibodies and cellular immune responses are urgently needed and important. To achieve this goal, rational vaccine design, including antigen modeling, screening and combination, vaccine pipelines, and delivery, are keys to developing a next-generation COVID-19 vaccine. In this study, we designed several DNA constructs based on codon-optimized spike coding regions of several SARS-CoV-2 variants and analyzed their cross-reactive antibodies, including neutralizing antibodies, and cellular immune responses against several VOCs in C57BL/6 mice. The results revealed that different SARS-CoV-2 VOCs induced different cross-reactivity; pBeta, a DNA vaccine encoding the spike protein of the Beta variant, elicited broader cross-reactive neutralizing antibodies against other variants including the Omicron variants BA.1 and BA.4/5. This result demonstrates that the spike antigen from the Beta variant potentially serves as one of the antigens for multivalent vaccine design and development against variants of SARS-CoV-2.

## 1. Introduction

SARS-CoV-2 is the pathogen of the COVID-19 pandemic. Prior to 10 February 2023, there were more than 755 million confirmed cases of COVID-19, leading to 6.83 million cumulative deaths globally [1]. Since SARS-CoV-2 was identified in December 2019 [2], and virus variants have emerged continuously due to virus mutations [3,4,5]. The evolution of SARS-CoV-2 variants with spike region mutations could enhance transmissibility and reduce neutralization activity, posing new challenges to COVID-19 prevention and treatment, especially for vaccination strategies [6,7,8,9,10,11]. The World Health Organization (WHO) classified five variants, B.1.1.7 (Alpha), P.1 (Gamma), B.1.351 (Beta), B.1.617.2 (Delta), and B.1.1.529 (Omicron) as variants of concern (VOC); meanwhile, the Omicron variant has several lineages including BA.1, BA.2, BA.3, BA.4, and BA.5 and their descendent lineages, and several circulating recombinant forms [5,12,13,14].

The limited and short-term immunity in COVID-19 convalescents is insufficient to protect against emerging variants [15,16]. Various COVID-19 vaccines available for human use are designed based on the wild-type of SARS-CoV-2, including the widely used mRNA vaccines BNT162b2 (Pfizer) [7,17] and mRNA-1273 (Moderna) [18,19] in Europe and North America, adenovirus vector-based vaccines ChAdOx-1 S [6,20], Ad26.COV2.S [21], Ad5-nCoV [22], recombinant subunit vaccine [23,24], inactivated vaccines [25,26], and DNA vaccines INO-4800 [27]. These vaccines elicited divergent cross-reactive T and B cell responses against different SARS-CoV-2 variants in COVID-19 vaccine recipients, especially neutralizing antibodies against the Omicron and Delta variants, which were substantially reduced to low levels, and this was responsible for breakthrough against infection by SARS-CoV-2 [28,29]. As noted, cross-reactive immune responses against different strains of SARS-CoV-2 are essential for the host to protect from continuously emerging variants. Cross-neutralizing antibody and T cell responses against SARS-CoV2 variants in COVID-19 patients and individuals immunized with different kinds of SARS-CoV-2 vaccines have been investigated in many studies and showed divergent and imbalanced neutralizing antibodies to each variant [30]. In contrast, cellular immune responses have broader a cross-reactivity due to the conservation of T cell epitopes among SARS-CoV-2 variants, even the Omicron variant [31].

Novel strategies are needed and being investigated to improve the efficacy of COVID-19 vaccines against emerging variants. Besides boosting doses with various vaccines originating from wild-type SARS-CoV-2 [32,33], vaccines composed of antigens from SARS-CoV-2 VOCs are being developed, and some have reached the clinical trial phase [34]. To elicit broader cross-reactive immunity covering most SARS-CoV-2 variants, a major issue is which viral variant should be chosen [35] and which immunization strategy is preferred [36,37].

To this end, we designed and constructed three DNA vaccines encoding the full-length spike protein of wild-type SARS-CoV-2 and two variants of Beta and Gamma. We evaluated their immunogenicity in C57BL/6, especially cross-neutralizing antibodies against major VOCs. Our results found that both Beta and Gamma variant DNA vaccines elicited strong cross-reactive humoral and cellular immune responses against wild-type SARS-CoV-2. In contrast, three DNA vaccines elicited divergent cross-neutralization for different variants, while pBeta elicited broader cross-reactive immune responses against other variants.

## 2. Materials and Methods

### 2.1. Animals

Female C57BL/6 mice aged 6–8 weeks old were purchased from Cavens Biogle Model Animal Research Co., Ltd. (Suzhou, China) and maintained in SPF conditions. The animal protocols were approved by IACUC of Advaccine (Suzhou) Biopharmaceutics Co., Ltd. (Suzhou, China), and the document number is 2021070102.

### 2.2. Design, Construction, and Identification of DNA Vaccines

pWT was constructed by inserting a full-length spike-coding gene of SARS-CoV-2 into the pVAX1 vector, which was named the pGX9501 described in the previous publication [38]. pWT (pGX9501) was proven to induce potent humoral, cellular immunity, and a protective effect after a wild-type SARS-CoV-2 challenge in hACE2 transgenic mice [39]. DNA vaccines against the SARS-CoV-2 P.1 (Gamma) variant and B.1.351 (Beta) variant were constructed using the same strategy. Genes encoding the spike of variant P.1 and B.1.351 were optimized using the same method according to human codon bias as previously indicated. Additionally, a unique signal peptide was added and followed by the spike gene. After restriction enzyme analysis and DNA sequencing, these DNA vaccines were transfected into HEK293T cells using the Hieff Trans^TM^ Liposomal Transfection Reagent (YEASEN, Shanghai, China), according to the manufacturer’s instructions. Cells were harvested at 48 h post transfection for Western blot analysis.

### 2.3. Western Blot

Cell lysates were prepared from the transfected cells with pWT, pBeta, and pGamma, and the cells transfected with the pVAX1 vector served as a negative control. The commercial S protein of wild-type SARS-CoV-2 (ACRO Biosystem, Beijing, China) served as a positive control. The primary antibody was an S-ECD/RBD monoclonal antibody (Bioworld, Nanjing, China), which is specifically recognized as the wild-type SARS-CoV-2 spike protein and has been shown to cross-react with the spike of other variants with a diverse binding affinity. The secondary antibody was HRP-conjugated anti-Human IgG (BD Biosciences, San Diego, CA, USA). ECL solution (BD Biosciences, San Diego, CA, USA) was used to visualize bands on the membrane.

### 2.4. Immunization

Female C57BL/6 mice at 6–8 weeks old were divided into four groups, and each mouse was either given pWT, pBeta, or pGamma, with pVAX1 serving as a negative control. A 25 μg DNA vaccine in 30 μL of SSC was injected intramuscularly, immediately followed by an electroporation device with Cellectra^®^2000 (Inovio, San Diego, CA, USA), which took place twice at the two-week interval. Mice were sacrificed on days 14 and 21 to evaluate humoral and cellular immune responses.

### 2.5. Binding Antibody Detection by ELISA

As described previously, ELISA detected the specific anti-RBD of SARS-CoV-2 wild-type binding antibody in the serum from each mouse [40]. The recombinant RBD of wild-type SARS-CoV-2 expressed in HEK293 cells (SinoBiological, Beijing, China) was used as coating antigen, whose final concentration was 1 μg/mL. Sera were three-fold serially diluted starting at 1:200, using PBST containing 3% BSA. The antibody titer was defined as the reciprocal of the largest serum dilution for which the OD_450nm_ value was greater than 2.1-fold of the negative sera, according to the previous method [39].

### 2.6. Neutralization Antibody Detection

The pseudo-virus microneutralization assay was performed to measure neutralizing antibody levels against wild-type SARS-CoV-2 and variants, as described previously [40]. Pseudovirus stocks of wild-type SARS-CoV-2 and VOC B.1.35, P.1, and B.1.617.2 were prepared at Fudan University, titrated, and aliquoted for storage at −80 °C. hACE2 stable expressing HEK293T cells (prepared in our lab) were used as target cells plated at 10,000 cells/well. Serum from each mouse was heat-inactivated and serially diluted 3-fold, starting at a 1:30 dilution. Sera were preincubated with SARS-CoV-2 pseudo-virus for 90 min at room temperature; then, the sera–pseudovirus mixture was added to hACE2-HEK293T cells and allowed to incubate in a standard incubator at 37% humidity, 5% CO_2_ for 72 h. After 72 h, cells were lysed using Bright-Glo™ Luciferase Assay (Promega Corporation, Madison, WI, USA), and RLU was measured using an automated luminometer. Neutralization titers (NT_50_) were calculated using GraphPad Prism 9.0 and defined as the reciprocal serum dilution at which RLU was reduced by 50% compared to RLU in virus control wells after the subtraction of background RLU in cell control wells.

### 2.7. ELISpot

ELISpot was performed using Mouse IFN-γ ELISpot PLUS plates (MABTECH, Cincinnati, OH, USA) according to the manufacturer’s protocol. Briefly, 2 × 10^5^ mouse splenocytes were plated into each well and stimulated for 20 h with peptide pools of 15-mer peptides synthesized by GenScript and overlapping by nine amino acids with the SARS-CoV-2. RPMI1640 containing 10% FCS and PMA/Iono were used for negative and positive controls, respectively. Spots were scanned and quantified using an AID ImmunoSpot reader (CTL, Cleveland, OH, USA), and IFN-γ spot-forming units were calculated and expressed as SFUs per million cells.

### 2.8. Intracellular Staining (ICS)

Intracellular cytokine staining of stimulated splenocytes was performed as in the previous method [40]. Splenocytes harvested from C57BL/6 mice were stimulated with the overlapping peptide pool of the wild-type SARS-CoV-2 spike protein for 6 h at 37 °C, with 5% CO_2_, stained with the following antibody cocktails CD4-FITC(0.5 mg/mL, 1:500 dilution)/CD8-PE-Cy5.5(0.2 mg/mL, 1:333 dilution)/IFN-γ-PE-Cy7(0.2 mg/mL, 1:125 dilution) /FVD-eFluor780 (1:1000) from eBioscience, and finally collected via Attune NxT Flow Cytometer (ThermoFisher, Waltham, MA, USA). Data were analyzed by FlowJo 10.0 software (TreeStar, Ashland, OR, USA).

### 2.9. In Vivo CTL Assay

Experiments were performed as described previously [39]. In brief, target cells were an equal mixture of 5 nM of eflour 450-labeled SARS-CoV-2 WT spike peptide pool-pulsed splenocytes and 0.5 nM of eflour 450-labeled un-pulsed splenocytes from naïve C57BL/6 mice. Mice from different groups on days 14 and 21 post-immunization received the target cells by tail vein injections, and cells were harvested after six hours. The percentage of labeled cells was detected with an Attune NxT Flow Cytometer and analyzed by FlowJo software. The specific cell lysis efficiency was calculated as follows: specific cell lysis efficiency (%) = (1 − (percentage of cells incubated with peptide/percentage of cells incubated without peptide)) × 100%.

### 2.10. Statistics

Statistical analyses were performed using GraphPad Prism 9.0 (GraphPad Software, San Diego, CA, USA). A non-parametric *t*-test was used to analyze the statistical difference for two-group experiments; one-way ANOVA was used to analyze multiple groups followed by Tukey’s post hoc test. It was regarded as a statistical difference when the *p*-value was less than 0.05.

## 3. Results

### 3.1. Design DNA Vaccine against Various SARS-CoV-2 Variants

The genes encoding the spike protein of SARS-CoV-2 wild type, two VOCs P.1 (Gamma) and B.1.351 (Beta) (Figure 1A) were optimized according to human codon bias, and a unique signal peptide was placed at the N-terminus of the spike protein. The recombinant plasmids pWT, pBeta, and pGamma were identified correctly via restriction enzyme (*BamH* I and *Xho* I) analysis (Figure 1B) and DNA sequencing. The expression of the spike protein of each construct was measured in HEK293T cells transfected with pWT, pBeta, and pGamma using Western blot (Figure 1C). The expression level of each construct appeared different, which could be explained by the diverse binding affinity of monoclonal antibodies to each variant of SARS-CoV-2.

### 3.2. DNA Vaccines pWT, pBeta, and pGamma Elicited High Binding Antibody Levels to Wild-Type SARS-CoV-2

After confirming the expression of each DNA vaccine in vitro, we next evaluated the immunogenicity of DNA vaccines in C57BL/6 mice via intramuscular immunization plus electroporation (Figure 2A). Receptor binding domain (RBD)-binding antibodies were measured via ELISA using the recombinant RBD of wild-type SARS-CoV-2 as the coating antigen in sera from each group on day 14 after prime and day 7 after boost (i.e., day 21 after prime). DNA vaccine pWT elicited a high titer of anti-RBD antibody IgG both on day 14 after the first immunization, reaching 3.13 × 10^4^ (Figure 2B), and elevated to 3.55 × 10^5^ on day 7 after the second immunization, which was 11-fold higher levels than that in mouse serum on day 14 (Figure 2C), implying the ability of pWT to trigger quick and robust antibody responses.

Using the same strategy, we constructed two DNA vaccines against two SARS-CoV-2 VOCs, P.1 and B.1.351, designated as pBeta and pGamma, respectively. Both DNA vaccines elicited high levels of anti-RBD antibody responses as pWT did, and it is gratifying to note that pBeta and pGamma induced high titers of cross-reactive antibody against the RBD of wild-type SARS-CoV-2, which reached 4.86 × 10^4^ and 4.86 × 10^4^, respectively, on day 14 (Figure 2B), which was quickly elevated to 4.04 × 10^5^ and 3.55 × 10^5^ on day 21 (Figure 2C). These results indicated that all three DNA vaccines had capacities to elicit quick and robust antibody responses, and each DNA vaccine against SARS-CoV-2 variants induced cross-reactive responses.

### 3.3. DNA Vaccines pWT, pBeta, and pGamma Elicited Robust Cellular Immune Responses

Next, we evaluated cellular immune responses elicited by DNA vaccines on day 14 after and day 21 (i.e., day 7 post-boost) after the first immunization using ELISpot and intracellular staining. Splenocytes from each mouse were stimulated with peptide pools of wild-type SARS-CoV-2 RBD (15 aa-length with 9-aa overlay). IFN-γ producing cells in splenocytes from pWT immunized mice on day 14 post-immunization were shown by ELISpot to have 311 SFU per million cells, significantly higher than that of pVAX1 control (*p* < 0.001) (Figure 3A), which increased to 1576 SFU per million cells after 7 days post-boost (Figure 3B). This result suggests that pWT triggers robust, specific cellular immune responses against the SARS-CoV-2 wild-type strain. The other two DNA vaccines, pBeta and pGamma (Figure 3A,B), also triggered a similar pattern of cellular immune responses in both immunized mice. Intracellular staining of IFN-γ in splenocytes stimulated with peptide pools showed that most of the IFN-γ-secreting cells detected in ELISpot were antigen-specific CD8^+^ T cells (Figure 3D and Figure 4C), and not CD4^+^ T cells (Figure 3C and Figure 4B). Because the stimuli peptide pool originated from the wild-type virus strain, the observed responses elicited by pBeta and pGamma were cross-reactive, indicating highly conserved cellular immunity among different SARS-CoV-2 variants.

We finally observed whether such a strong cellular immune response was sufficient to have a cytotoxic function in vivo. An in vivo killing assay using eflour 450-labeled cells showed that all three DNA vaccines, pWT, pBeta, and pGamma, were able to generate a highly efficient specific killing activity of SARS-CoV-2 wild type. On day 14 post-immunization, pWT, pBeta, and pGamma induced high levels of WT-specific killing efficiency, reaching 88.67%, 86.67%, and 63.67%, respectively (Figure 5A), and remaining at similar levels on day 21 post-vaccination (Figure 5B). These results further demonstrated that DNA vaccines induced robust, cross-reactive, and functional cellular immune responses.

### 3.4. DNA Vaccines Induced the Highest Neutralizing Antibody for Each Variant, but Diverse Cross-Reactive Neutralizing Antibodies for Other VOCs

All three DNA vaccines induced high titers of neutralizing antibodies for each variant. The geometric mean value of pVNT_50_ measured via pseudo-typed neutralization assay were 5923, 5901, and 4010, respectively, against the wild-type of SARS-CoV-2 (Figure 6A), and Beta (Figure 6B) and Gamma (Figure 6C) variants.

With the increasing emergence of SARS-CoV-2 variants, the cross-reactive neutralizing antibody induced by each virus variant becomes a more critical issue for COVID-19 vaccines. We next detected neutralizing antibodies against several VOCs elicited by pWT, and found that pWT only induced high titers of neutralizing antibodies for wild-type SARS- CoV-2 (Figure 6A), low to moderate levels of neutralizing antibodies for SARS-CoV-2 variant Beta (Figure 7A), Gamma (Figure 7B) and Delta (Figure 7C), but no detectable NtAbs for Omicron variants BA.1 (Figure 7D) and BA.4/5 (Figure 7E).

Then, we denoted neutralizing antibody titer against different variants elicited by pWT as 1.00. Antibodies elicited by pBeta and pGamma were divided by NtAb induced by pWT to show the cross-reactivity of each DNA vaccine against different variants. The results showed that diverse antibody profiles were induced by three DNA vaccines (Figure 7F). pWT, a DNA vaccine containing a gene-encoding wild-type spike, elicited less cross-neutralizing antibody against other SARS-CoV-2 VOCs, as described above. DNA vaccine pBeta induced relatively broader neutralizing antibody against other VOCs, whereas antibody titers against the wild type were 1.43 times that of pWT. NtAb to variants Beta, Gamma, Delta, and Omicron BA.1 and BA.4/5 were 26.78, 10.97, 4.32, and 6.17 times to that of pWT, respectively. pGamma also showed better cross-neutralizing ability than pWT did, but its cross-reactivity was lower than pBeta. These results demonstrated that the SARS-CoV-2 Omicron BA.4/5 variant has the lowest cross reactivity with the wild-type and other variants, which is consistent with others’ findings, and the Beta variant has a relatively broader cross reactivity.

In general, DNA vaccines against wild-type SARS-CoV-2, the Beta variant, and the Gamma variant immunized by electroporation had good immunogenicity to trigger robust humoral and cellular immune responses. Each DNA vaccine elicited a diverse pattern of cross-reactive immune responses, providing quick and efficient information to design a universal vaccine against the increasing emergence of SARS-CoV-2 variants. Due to its high efficiency, low-cost, and easy storage and transportation, the DNA vaccine may be a potential vaccine against SARS-CoV-2 and serve as a convenient platform to screen effective antigen candidates for other vaccine development platforms.

## 4. Discussion

With the increasing emergence of SARS-CoV-2 variants, it becomes more and more important whether a SARS-CoV-2 vaccine confers cross-reactive protection against emerging virus variants. Present investigations on cross-reactive immune response and protection in animal models and humans (including clinical trials and real-world observation) provide valuable information. So far, various COVID-19 vaccines available for human use are designed based on the original SARS-CoV-2. Several studies investigated their efficacy against various emerging variants, especially VOCs; strategies are also attempted to obtain better and broader protective immunity. Besides boosters with homologous vaccines, heterologous sequential immunization and universal vaccine development are two major strategies. Heterologous sequential immunization, either with vaccines originating from wild-type SARS-CoV-2 by different platforms or with some vaccines consisting of antigens from different emerging VOCs, may confer better protection from various existing and emerging virus variants. However, one major issue for boosting vaccination and universal vaccine development is which virus variant should be chosen. There is also the question of what the vaccine pipeline is prior to selection.

DNA vaccines were studied in the 1990s, when they did not have the capacity of inducing high levels of humoral immune responses; with technological progress, their ability to elicit humoral immune responses has been greatly improved in recent years. Unlike the mRNA vaccine, the DNA vaccine is easy to produce and transport to remote areas due to their stability and low cost, indicating the advantages of DNA vaccines over other forms of vaccines to be applied in the future. In November 2021, a DNA vaccine candidate against SARS-CoV-2, ZyCoV-D was the first approved for emergency use in India. Several others are also being developed and in different phases of clinical trials, including INO-4800 (phase 2/3, Advaccine/Inovio), AG0301-COVID19 (Phase 2/3-AnGes), and Covigenix VAX-001 (Phase1, Entos Pharmaceuticals Inc., Edmonton, AB, Canada).

In this study, we constructed three DNA vaccine candidates, pWT, pBeta, and pGamma, encoding the full-length spike protein of wild-type SARS-CoV-2, the Beta variant, and Gamma variant. To achieve potent immunogenicity, codon optimizations, and unique signal peptides at the N-terminus of protein were introduced in those vaccine designs. Expression identification in vitro and immunogenicity in vivo demonstrated the efficiency of our design strategy. We noticed that the expression level of each construct by Western blot assay was inconsistent with the magnitude of antibody and cellular immune responses, likely because the band intensity (Figure 1C) on the Western blot membrane did not accurately reflect the real expression level of each construct. In our Western blot assay, S-ECD/RBD monoclonal antibody was used as a primary antibody, which has a different affinity to the spike protein from other variants of SARS-CoV-2.

We first evaluated antibody and cellular immune responses based on constructing three DNA vaccine candidates. pWT, a DNA vaccine candidate previously made by our group, namely pGX9501, was proven to induce effective humoral and cellular immune responses against live viral challenge infection with original SARS-CoV-2 [40] [Human Vaccine]. Consistent with our previous study, pWT immunized in C57BL/6 mice induced quick and high levels of antibody and T cell responses in this study. Interestingly, pBeta and pGamma induced robust cross-reactive humoral and cellular immune responses against wild-type SARS-CoV-2. pBeta and pGamma elicited high titers of binding antibodies against wild-type SARS-CoV-2 RBD, albeit with a slightly decreased trend for cross-neutralizing antibodies (without statistical significance). A similar magnitude of cellular immune responses was observed with ELISpot, most of which was CTL response, as further demonstrated by FACS analysis and the in vivo killing assay. These results are consistent with previous reports that cellular immune responses have broader cross-reactivity due to T cell epitope conservation among SARS-CoV-2 variants, even against Omicron variants, suggesting that a vaccine eliciting cross-reactive cellular immune response could be developed as a universal vaccine against SARS-CoV-2.

The capacity of a vaccine that induces cross-neutralizing antibodies for various SARS-CoV-2 variants is the most important for future COVID-19 vaccine development. Recently, many researchers focused on cross-neutralizing antibody responses from current vaccines and natural infections. Compared to the broader cross-reactivity of cellular immune responses, divergent and imbalanced neutralizing antibodies for each variant were observed in recovered COVID-19 patients and vaccinated individuals. In most findings, a cross-neutralizing antibody against the Omicron variant was the lowest, followed by the Delta variant in convalescent patients with wild-type SARS-CoV-2, and other variants and individuals vaccinated with vaccines available in use. Controversial results were also observed in several studies. The sera of convalescent patients infected with the wild-type strain maintained a higher level of cross-neutralizing antibody titers than those infected with the other VOCs [41]. Plasma or sera from individuals infected with the original SARS-CoV-2 and mRNA vaccines showed substantially lower neutralization to Beta variant [42], and patients infected with the Beta variant generated low levels of the cross-neutralizing antibody for the wild-type virus (only one-third of Beta) and high levels of neutralizing antibody for the Alpha variant, which might be due to the vastly different N-terminal domains of these variants [30].

Our results demonstrated that the DNA vaccine pWT induced fewer cross-neutralizing antibodies against Beta, Gamma, Delta, and no detectable neutralizing antibodies against the Omicron BA.1 and BA.4/5 variants. In contrast, pBeta elicited a relatively broader range of cross-neutralizing antibodies, nearly equivalent to the neutralizing antibodies against wild-type SARS-CoV-2 and the Gamma variant, a moderate level of neutralizing antibody against the Delta variant, and even a low but detectable neutralizing antibody level against Omicron BA.1 and BA.4/5, which could be explained by mutation-caused conformational structure changes. By comprehensively considering cross-reactive neutralizing antibody and cellular immune responses, pBeta may serve as a vaccine candidate for the heterologous sequential immunization of people who have previously received other SARS-CoV-2 vaccines, such as an inactivated vaccine, mRNA, or recombinant adenovirus; on the other hand, the Beta variant spike protein may be selected as an antigen candidate to design universal COVID-19 vaccines.

Although our findings on cross-reactive immune responses are obtained from DNA vaccines, they may be generalizable to other kinds of vaccines, including RNA vaccines, recombinant vector-based vaccines, recombinant subunit protein vaccines, and inactivated vaccines. The major limitation of this study is that the DNA vaccines encoding Omicron variants were not included, which may elicit higher titers of neutralizing antibodies against the current dominant Omicron strains. It has proven that Omicron-containing mRNA vaccines elicited more neutralizing antibody responses against Omicron variants [43]; however, a phase 2/3 clinical trial initiated by Moderna, Inc. demonstrated that the mRNA-1273.211 vaccine (containing equal mRNA amounts encoding the ancestral SARS-CoV-2 and Beta variant spike proteins) elicited robust and long-term antibody responses against multiple SARS-CoV-2 VOCs [44].

In a subsequent study, we will investigate cross-reactive immune responses elicited by DNA vaccine constructs encoding currently epidemic strains and new emerging strains to provide more systemic information for vaccine development. Additionally, DNA vaccines may also serve as a platform to screen optimal antigens for vaccine design, not only suitable for SARS-CoV-2 but also suitable for other pathogens and noninfectious diseases in the future.

## Figures and Tables

**Figure 1 vaccines-11-00513-f001:**
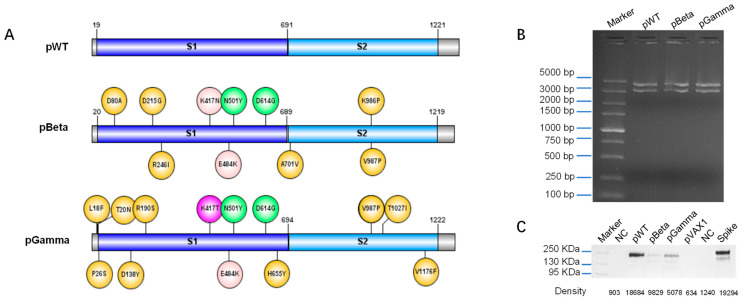
Design, construction, and identification of DNA vaccine against various SARS-CoV-2 variants. (**A**) Spike protein of the wild type, Beta and Gamma variants of SARS-CoV-2 for DNA vaccine constructs, (**B**) restriction enzyme analysis of each DNA construct using *BamH* I and *Xho* I, with DL5000 DNA Marker (Takara) was used as a DNA marker, (**C**) Western blot analysis of cell lysates from plasmid-transfected HEK-293 cells over 48 h, with PageRuler™ Plus Prestained Protein Ladder 10–250 kDa used as a protein marker. “NC” represents negative control which was the cells transfected with the pVAX1 vector, and “Spike” refers to the commercial spike protein of wild-type SARS-CoV-2 (ACRO Biosystem, Beijing, China) which served as the positive control.

**Figure 2 vaccines-11-00513-f002:**
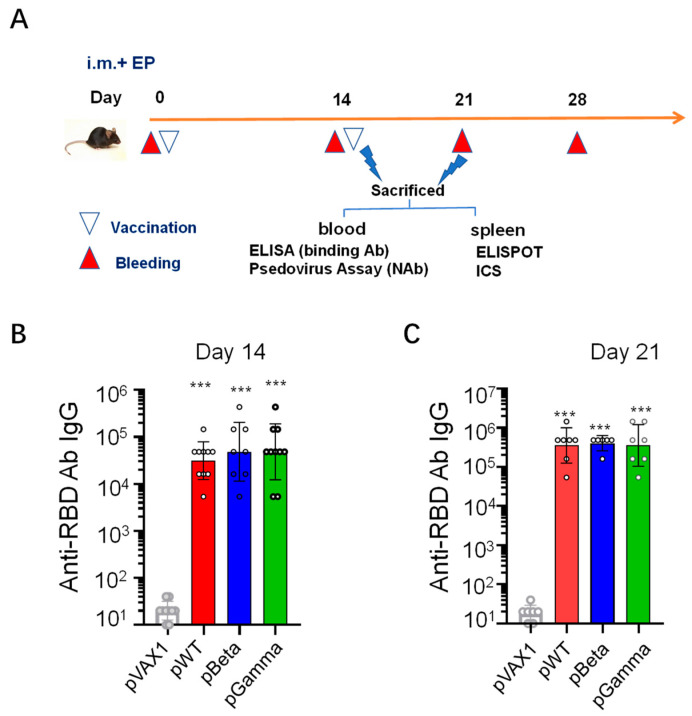
Vaccination schedule and binding antibody against RBD in sera from DNA vaccine immunized C57BL/6 mice. (**A**) Mice were intramuscularly immunized twice at two-week intervals with DNA vaccine at a dose of 20 μg in 50 μL PBS at the quadriceps following electroporation, and sacrificed on days 14 and 21. Titers of anti-RBD specific IgG antibodies were measured using ELISA and expressed as geometric mean ± SD, (**B**) RBD-binding antibody titer elicited by pWT (*n* = 10), pBeta (*n* = 8) and pGamma (*n* = 10) on day 14, (**C**) RBD-binding antibody titer elicited by pWT (*n* = 7), pBeta (*n* = 6) and pGamma (*n* = 7) on day 21. *** denotes extremely significant difference (*p* < 0.001) compared with pVAX1.

**Figure 3 vaccines-11-00513-f003:**
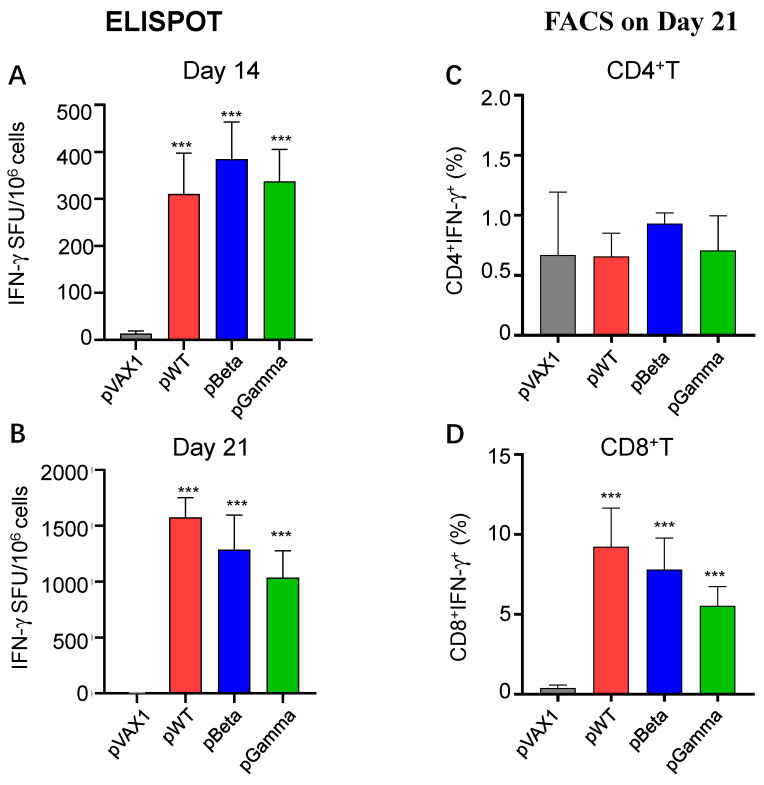
ELISPOT and FACS analysis of splenocytes from mice immunized with various DNA vaccines. Splenocytes were harvested on days 14 and 21 post-immunization and stimulated with an RBD peptide pool of wild-type SARS-CoV-2. ELISPOT was performed to observe IFNγ-secreting cells on days 14 and 21. (**A**) Spot forming unit (SFU) per million splenocytes of pWT, pBeta, and pGamma on day 14 (*n* = 3), (**B**) SFU per million splenocytes on day 21 (*n* = 3). (**C**) A percentage of IFNγ^+^CD4^+^ T cells in vaccinated mice on day 21 by FACS, (**D**) a percentage of IFNγ^+^CD4^+^ T cells in vaccinated mice on day 21 by FACS. *** denotes a significant difference compared with pVAX1 (*p* < 0.001).

**Figure 4 vaccines-11-00513-f004:**
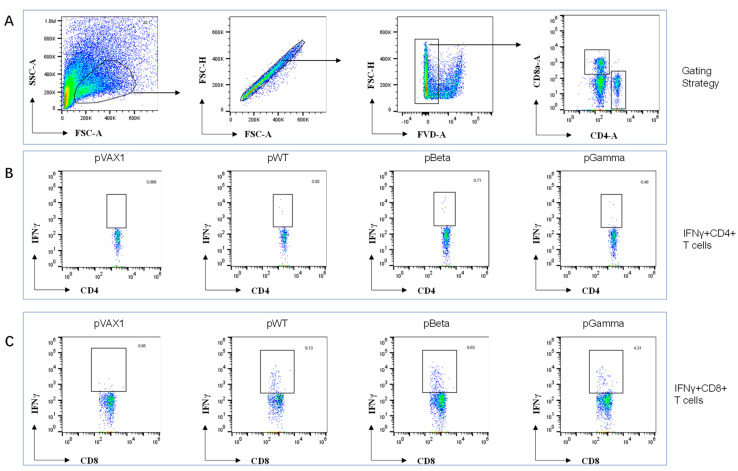
FACS analysis of splenocytes from mice immunized with various DNA vaccine constructs on day 21. (**A**) Gating strategy of CD4 and CD8 T cells from splenocytes, (**B**) representative data of IFNγ^+^CD4^+^ T cells, and (**C**) IFNγ^+^CD8^+^ T cells in spleens from mice immunized with the four DNA vaccine constructs: pVAX1, pWT, pBeta and pGamma.

**Figure 5 vaccines-11-00513-f005:**
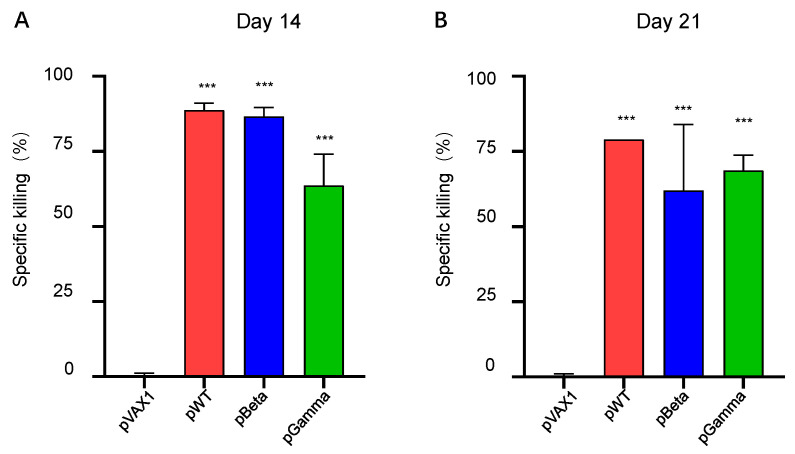
In vivo killing assay on day 21 post-immunization. An equal amount of 5 nM eflour 450-labeled SARS-COV-2 WT spike RBD-originated peptide-pool-pulsed splenocytes and 0.5 nM eflour 450-labeled un-pulsed splenocytes from naïve C57BL/6 mice were used as target cells and adoptively transferred to mice from each group. The specific killing percentage was calculated as described in the Section 2
. (**A**) Specific killing percentage on day 14, (**B**) specific killing percentage on day 21 by DNA vaccine pWT, pBeta, and pGamma (*n* = 3). *** denotes a significant difference compared with pVAX1 (*p* < 0.001).

**Figure 6 vaccines-11-00513-f006:**
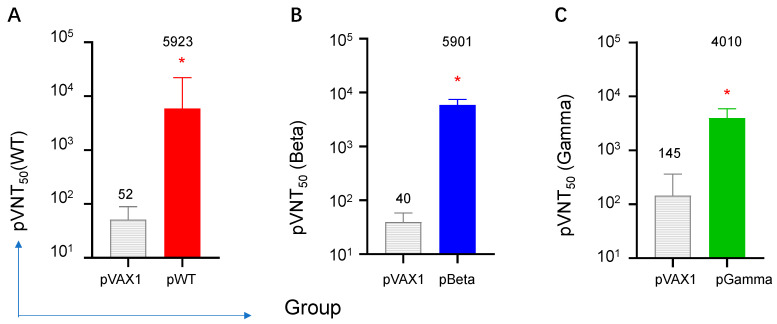
Neutralizing antibodies in mouse sera were detected via pseudo-virus neutralization on day 21. Sera from immunized mice were three-fold serially diluted starting at 1:30. Neutralization titers were calculated using GraphPad Prism 9.0 and defined as the reciprocal serum dilution at which RLU was reduced by 50% compared to virus control and expressed as geometric mean ± SD. (**A**) Serum pVNT_50_ to wild type SARS-CoV-2 by pWT (*n* = 4), (**B**) serum pVNT_50_ to SARS-CoV-2 variant Beta by pBeta (*n* = 4), (**C**) serum pVNT_50_ to SARS-CoV-2 variant Gamma by pGamma (*n* = 4). * denotes a significant difference compared with pVAX1 (*p* < 0.05).

**Figure 7 vaccines-11-00513-f007:**
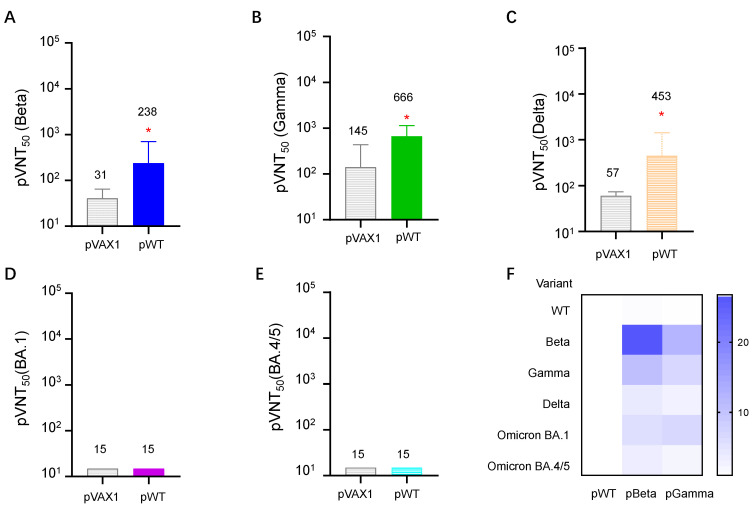
Cross-reactive neutralizing antibodies in mouse sera were detected by pseudo-virus neutralization on day 21. Sera from pWT-immunized mice were three-fold serially diluted starting at 1:30. Neutralization titers were calculated using GraphPad Prism 9.0 and defined as the reciprocal serum dilution at which RLU was reduced by 50% relative to virus control, and expressed as geometric mean ± SD. (**A**) Serum pVNT_50_ to SARS-CoV-2 Beta variant, (**B**) serum pVNT_50_ to SARS-CoV-2 variant Gamma, (**C**) serum pVNT_50_ to SARS-CoV-2 variant Delta, (**D**) serum pVNT_50_ to SARS-CoV-2 variant Omicron BA.1, (**E**) serum pVNT_50_ to SARS-CoV-2 variant Omicron BA.4/5, (**F**) heatmap of relative neutralizing antibody elicited by pWT, pBeta, and pGamma. pVNT_50_ of each virus variant elicited by pWT was denoted as 1.00. * denotes a significant difference compared with pVAX1 (*p* < 0.05).

## Data Availability

Data will be made available upon request to the corresponding author though may be subject to a Material Transfer Agreement between institutions.

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
