# Peer review of "A DNA Vaccine Encoding the Full-Length Spike Protein of Beta Variant (B.1.351) Elicited Broader Cross-Reactive Immune Responses against Other SARS-CoV-2 Variants"

_vaccines, 2023, doi:10.3390/vaccines11030513_

Round 1

Reviewer 1 Report (Previous Reviewer 2)

The authors have addressed the previous review comments point by point and modified the main text accordingly.

It’s a pity that the authors didn’t apply the DNA vaccine strategy to the Omicron variants, although the authors mentioned in their response letter that their major aim is to use DNA vaccine as a rapid and effective platform to screen antigen candidate for future vaccine development, with the purpose of eliciting broad cross-reactive immune responses against emerging variants. The authors may need to emphasize this in their discussion section, and may mention if the DNA vaccine is designed based on Omicron variants they may elicit more neutralizing Ab responses against the strains that are dominant currently.

In Figure 5C and Figure 6C, could the authors explain why the negative control group of pVAX1 showed increased neutralization titer against the pseudovirus of Gamma variant? Is it due to virus titer itself?

In Table 1, it may be clearer if the authors can make the table in a heatmap format (i.e. have the background of each cell as a gradient of color to distinguish the relative neutralization values) since the numbers themselves are not obvious enough.

Author Response

Dear Professor, the atteched file is the point-to-ponit responses to your comments.Accordingly, each comment was addressed in the revised manuscript.

Reviewer 2 Report (Previous Reviewer 3)

See attached file

Author Response

Dear Professor, the attached file is the point-to-point responses to your comments.Accodingly, each comment and suggestion were addressed in the revised manuscript.

Reviewer 3 Report (Previous Reviewer 1)

-Comment: While I understand that Beta spike protein could generate antibodies that cross react with newer Omicron antigens and that your assay could be used for diagnostic purposes, how is your work relevant to the current situation?  Omicron is the predominant circulating strain, with some sub-lineages / recombinants, and Beta or Gamma are no longer relevant for future vaccination. therefore using Beta in a multivalent vaccine is unlikely. In Europe, these are actually not considered as VOC anymore. A booster (Omicron BA5) is available in the US since October 2022. Please include in the discussion the limitations of your work regarding that aspect. 

-What is the reasoning for this particular mouse model when the references you cite are from studies using humanized ACE2 mouse models? The choice of the model needs to be justified in the M&M.

-Confusion regarding viruses used throughout the study can emerge when you are referring to wild-type, prototype, and ancestral. this prevents from proper understanding of what was done and analysis. Referencing to a study that used either one does not excuse you form detailing what virus was used. You need to add that information everywhere you reference to either term for the reviewer to follow your work. 

-Figure 6D is Delta not BA.1. 

-Editing for proper use of English is required. Also, please add a period instead of a comma where needed.

-No need for Appendix A

Author Response

Dear Professor, the attached file is the point-to-point responses to your comments.Accodingly, each comment and suggestion were addressed in the revised manuscript.

Round 2

Reviewer 2 Report (Previous Reviewer 3)

The manuscript is improved.

Reviewer 3 Report (Previous Reviewer 1)

Manuscrit still needs to be edited for proper English. There are missing words and typos. Use of the wrong tense was also noted several times throughout the manuscript.

This manuscript is a resubmission of an earlier submission. The following is a list of the peer review reports and author responses from that submission.

Round 1

Reviewer 1 Report

Zhao et al constructed 3 DNA vaccines encoding the full spike protein of the prototype SARS-CoV-2 and 2 VOC (Beta and Gamma), and then tested their respective immunogenicity in C57BL/6 mice. There is a clear need in developing better vaccines and the manuscript is of interest. However, this is not acceptable in the present form as it requires extensive editing, especially in M&M and result section, for clarification and proper use of English. This makes it extremely difficult to navigate through the manuscript and understand what was done.  

a few specific comments:

-add in title B.1.351

-line 19: you mean codon optimized

-add proper punctuation throughout the manuscript

-rephrase 22-24, 34, 51, 68, 72-73, 106, 122-126. There is more to do.

-ref line 33 is not formatted

-in 2.2: please detail the origin of spike, the isolate and the sequence reference that is published. Also, be consistent the way you identify it: prototype or ancestral?

-2.5: please detail why 2.1 is your cut-off.

-Where is the peptide pool of 15-mer coming from?

-Why is P1 mentioned in the M&M is not used?

-Figure 1B are the restriction enzyme sites specific to each construct? if no how do you know you used the right construct in animals?

-Figure 1c is the pWT construct coding for the same antigen that is detected in the last lane (spike)?

Reviewer 2 Report

In this manuscript, Zhao et al. evaluated the DNA vaccine platform encoding the full-length spike proteins of different SARS-CoV-2 variants, including wild-type, Beta, and Gamma, for their ability to induce both cellular and humoral immune responses. The authors showed the DNA vaccine encoding Beta variant spike elicited relatively broader cross-reactivities against other VOCs, and suggested Beta variant as a template for future vaccine design. The manuscript was written in a clear manner; however, the following points will need to be addressed before its being considered acceptance to the journal.

Major points:

In this study, the authors evaluated DNA vaccines encoding the spike protein of prototype SARS-CoV-2 and two VOCs including Beta and Gamma. I’m wondering why the authors didn’t try the DNA vaccine encoding Omicron or Delta variants? Given the fact that the Omicron is most prevalent variant now, as well as the Delta variant which was previous prevalent.

Here the authors tested the vaccines in wild-type C57BL/6 mice, however, as also mentioned by the authors in line 85, the DNA vaccine encoding wild-type spike protein have been studied in hACE2 transgenic mice for protective effect against SARS-CoV-2 challenge, thus such viral challenge animal model may also be applied in this study to show the advantage of the vaccine encoding spike of Beta variant.

Quantification of protein expression by different DNA vaccine constructs may need to be improved. Instead of running western blot using primary antibody with different binding affinity to different VOCs, for example, the authors may use primary antibodies that bind to conserved region other than RBD which mutates heavily between VOCs. An alternative way may be directly measuring expression level of total protein yield after purification from the same expression volume.

The authors tested the antibodies from mice serum for their binding specific to SARS-CoV-2 RBD by ELISA, however, there are also antibodies targeting other domains of the spike protein other than RBD, for example, NTD or S2 epitopes. Measuring only RBD-specific antibody may not reflect the whole picture of the antibody response induced by the vaccines.

The authors performed ELISpot to test the mouse IFN-γ secretion to reflect T cell responses. Mice splenocytes were stimulated with peptide pools of only RBD of SARS-CoV-2, however, peptides from other regions outside RBD should also be considered. In addition, other cytokines may also be tested, for example, IL-2, which has commercial kits for mice available.

The authors need to explain in more details how the values of relative neutralization in Table 1 were calculated. Is it based on NT50 or other parameters? In addition, the neutralization results showed in Table 1 should also include other VOCs such as alpha, omicron (including BA.1, BA.2, BA.4/5, etc.) which can better show the real cross-reactive neutralizing antibody responses induced by these different vaccines.

Minor points:

The number of infected populations need to be updated in the first lines in introduction section.

In line 38, the order of the VOCs may need to be changed, from “alpha, gamma, beta, …” to “alpha, beta, gamma, …”.

In line 47, the authors may add the reference to Novavax vaccine for the recombinant subunit vaccine.

There is a typo in line 177, “SRAS-CoV-2” should be “SARS-CoV-2”.

Between line 196 and 197, “day 7” should be “day 14”.

In Figure 2, B,C,D can be combined as one plot, while E,F,G can be combined as one plot. Same for Figure 3.

In line 212, the first appearance of “ICS” needs its full name.

In line 214, before “15 aa-length with 9-aa overlay”, the left parenthesis is missing.

“beta” should be “Beta” with the first letter capitalized, in line 254 and line 318.

Reviewer 3 Report

Zhao et al. described the cross-reactivity potential of different DNA vaccines. They analyzed the immunization in mice and measured antibody binding, immune cell reaction and in vivo killing. Furthermore, they investigated neutralizing antibodies and cross-reactive neutralization.

The big issue with these data is the investigation of the SARS-CoV-2 variants beta and gamma while now omicron is predominant, and probably beta and gamma variants will not be relevant for future vaccination. Additionally, if there is a higher cross-reactivity in mice, it is not clear if this would be the same in human. Therefore, the relevance of this study is questionable.

Minor concerns

Introduction:

L64-66: You wrote about immunization strategies but never discuss this further.

Materials and Methods

L144-149: Used antibodies are missing (company, concentration/dilution, fluorochrome)

L161-164: Did you use a post hoc test for ANOVA?

Results

L168-169: “unique signal peptide” Which and why?

Figure 1: Figure Legend: define “NC” and “spike” in the caption

L186: “RBD”: define

Figure 2: Is pVAX1 in each graph the same? It would be easier for comparison if you put all variants in one graph.

L211: “ICS”: define

L212-217: structure of the sentence, rephrase

Figure 3: Again, it would be helpful to have all variants in one graph for comparison. Additionally, show primary FACS data (dot plots) for clarity.

L237-238: “kept at similar levels” This is not the truth because especially for pBeta the killing is reduced after d21 compared to d14 (50% vs 80%). Or is this a problem with numbers on axis? Only 5 numbers and 6 ticks.
